# Calibration and Verification Test of Cinnamon Soil Simulation Parameters Based on Discrete Element Method

Yiqing Qiu [1], Zhijun Guo [1,*], Xin Jin [2], Pangang Zhang [1], Shengjie Si [1] and Fugui Guo [1]

1   College of Vehicle and Traffic Engineering, Henan University of Science and Technology,
    Luoyang 471003, China; 195400000014@stu.haust.edu.cn (Y.Q.); 190319030213@stu.haust.edu.cn (P.Z.);
    200320030285@stu.haust.edu.cn (S.S.); 200320030333@stu.haust.edu.cn (F.G.)
2   College of Agricultural Engineering, Henan University of Science and Technology, Luoyang 471003, China;
    jx.771@haust.edu.cn
*   Correspondence: gzhj1970@haust.edu.cn

**Abstract:** To obtain the discrete element simulation model parameters suitable for the interaction between cinnamon soil and soil-engaging components, the Hertz–Mindlin with JKR contact model in EDEM simulation software was used to calibrate the relevant model parameters of cinnamon soil. Firstly, the particle size distribution, moisture content, volume density, Poisson's ratio, shear modulus, and other parameters of the cinnamon soil were measured with cinnamon soil as the research object. Further, taking the stacking angle as the response value, the Plackett–Burman test, the steepest climbing test, and the Box–Behnken were designed by using the Design-Expert software to calibrate and optimize the physical parameters of soil simulation. The optimal parameter combination was obtained: cinnamon soil–cinnamon soil rolling friction coefficient was 0.08, soil JKR surface energy was 0.37 J/m$^{-2}$, and cinnamon soil–steel static friction coefficient was 0.64. Finally, the discrete element simulation verification test of stacking angle and cutting resistance was carried out under the calibrated parameters. The comparative calculation showed that the relative error between the simulated stacking angle and the measured stacking angle was 0.253%, and the maximum relative error between the simulated cutting resistance and the measured cutting resistance was 10.32%, which was within the acceptable range, indicating the high accuracy and reliability for the calibration parameters. The research results have important reference value for the energy-saving and consumption-reducing design of soil tillage components and provide basic data for the simulation of cutting resistance research of cinnamon-soil-engaging components.

**Keywords:** cinnamon soil; discrete element method; parameter calibration; stacking angle; JKR contact model; bulldozing plate; cutting resistance

## 1. Introduction

Cinnamon soil is a zonal soil in the warm temperate arid forest or shrub grassland areas. Compared with ordinary soil, it has stronger expansibility and flow plasticity, having the characteristics of heavy texture and deep soil layer [1]. When farming or cutting cinnamon soil, problems such as high resistance, high energy consumption, and severe soil adhesion of soil-engaging components will occur, which will seriously affect the operation efficiency [2]. Thus, exploring the general principle of the soil cutting dynamics and the design method of the soil cutting components has become an urgent task, one that has attracted our attention. The rapid development of modern bionic science and technology provides insight and impetus for further research and solutions to this problem. Long-term evolution has created many animals with clawed toes in nature, such as voles, house mice, mole crickets, and chooks; these claws have excellent soil digging functions. It is a starting point to realize the effect of reducing the resistance and saving energy through the bionic design method of the macroscopically geometric structure of relevant soil cutting tools. When designing and optimizing critical soil-engaging components, it is often required to

study the interaction law between cinnamon soil and soil-engaging components. However, the actual soil bin test or field test can only understand the macroscopic movement of soil but cannot explore the microscopic movement law of the soil. At present, domestic (Luoyang, China) and foreign experts all use the finite element method to study the soil dynamic disturbance behavior during farming or cutting [3,4]. However, the finite element method is based on the expression form of a continuous medium to simulate the overall destruction behavior of soil, which cannot simulate the soil layer separation, cracking, and soil particle flow in the actual operation process [5]. In recent years, Cundall et al. [6] proposed a new numerical method, namely, the discrete element method, which breaks the limitation of treating soil as a continuum and is an ideal method for analyzing large deformation of discontinuous soil [7–9].

When the agricultural soil tillage components are farming or cutting operations in the field, the interaction between soil and soil-engaging components will also be different due to different soil types [10]. Therefore, when establishing discrete element modeling for different soils, an appropriate soil contact model should be selected, and relevant model parameters should be calibrated accurately according to the specific conditions of the soil to ensure the authenticity of the discrete element simulation test [11]. Ucgul et al. [12] compared the simulation results of nonlinear elastic and linear hysteretic spring (plastic) contact models with the test results of scanning tillage tools and obtained the variation law of soil physical contact parameters. Shi Linrong et al. [13] established the soil particle contact model for farmland in the arid region of Northwest China and calibrated the critical parameters by the discrete element method. Zhang Rui et al. [14] proposed a way to calibrate the simulation parameters of sand particles. Xing Jiejie et al. [15] completed the simulation parameter calibration of Hainan laterite in China by soil accumulation test, and the soil breaking test verified the effectiveness of the calibration parameters. Li Junwei et al. [16] used the JKR model in EDEM to calibrate the contact parameters of black soil particles and their contact parameters with soil-engaging components in Northeast China, which provided a reference for the calibration of high-water content clay parameters. The above research on soil parameter calibration mainly aims at soil types such as sandy soil, red soil, black soil, dry soil, and other soil types [17]. However, the research on the calibration of the simulation parameter system for cinnamon soil is rarely reported.

Therefore, aiming at the problem that the parameters of the discrete element model of cinnamon soil are not clear, this paper took the cinnamon soil with yellow sandy soil characteristics commonly found in wheat fields in Luoyang, China, as the research object, selected Hertz–Mindlin with JKR Cohesion model as the contact model, and calibrated the parameters of the discrete element model of cinnamon soil by the combination of physical test and simulation test. Applying comparative analysis of soil cutting resistance in a simulation test and indoor soil bin test, the reliability of the parameters of the simulation model was verified. We aim to provide basic data for the further development of soil-engaging components suitable for cinnamon soil conditions.

## 2. Determination of Intrinsic Soil Parameters

### 2.1. Soil Prototype and Particle Size Distribution

This work focuses on the relative controllable soil conditions of indoor soil bins. The soil in the soil bin was cinnamon soil with yellow sandy soil characteristics, which was taken from a wheat field in Luoyang, China (Figure 1). In order to reflect the actual working soil condition more accurately, considering the difference between the measured value of soil moisture content after rain and the soil moisture content under continuous sunny days, this paper used the configured cinnamon soil with an average moisture content of 11.62% as the object to calibrate the parameters of the discrete element soil particle contact model. The average soil bulk density was 1394 kg/m$^3$, and the soil moisture content was 11.62% measured by the oven method. The grading sieve was used to classify the soil particles, and the corresponding masses of different particle sizes were weighed by an electronic

scale. The corresponding mass fractions of particle sizes were calculated, and the results are shown in Table 1.

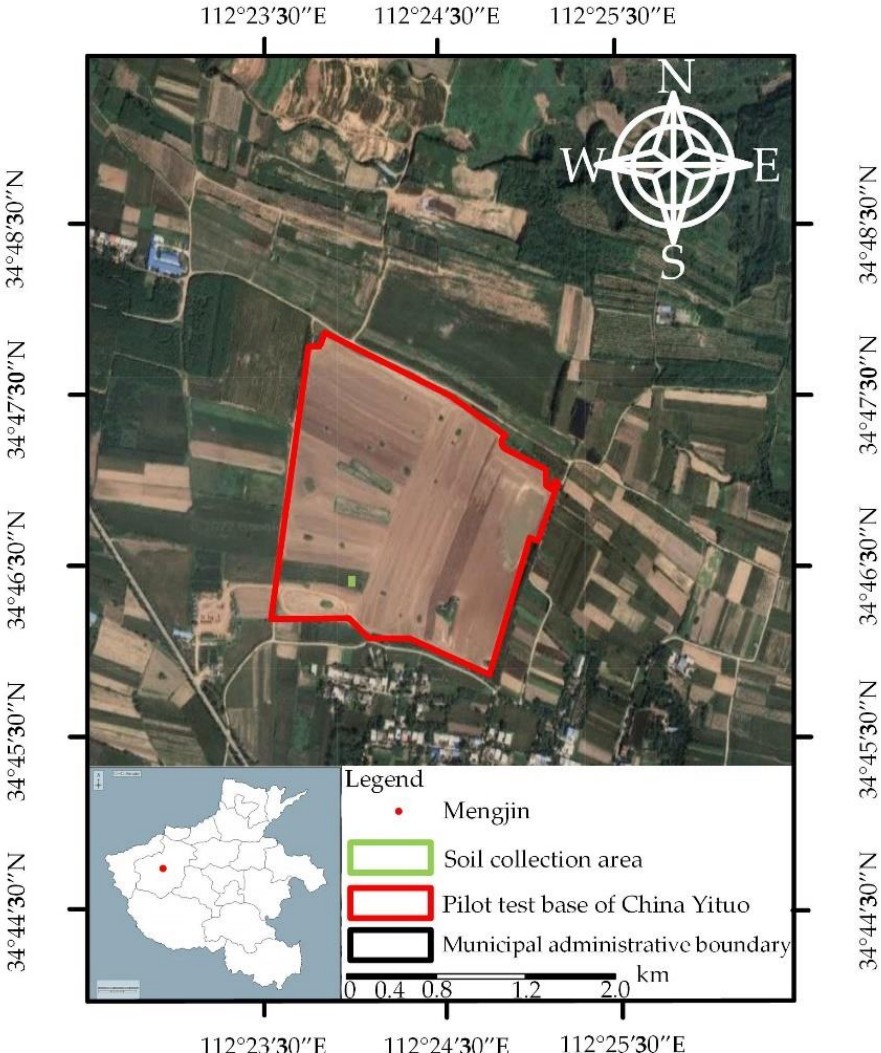

**Figure 1.** Soil sample site.

**Table 1.** Particle size distribution.

| Number | Particle Size Range/mm | Mass Fraction/% |
|--------|------------------------|-----------------|
| 1 | >2 | 8.85 |
| 2 | 1–2 | 8.20 |
| 3 | 0.5–1 | 17.50 |
| 4 | 0.25–0.5 | 17.66 |
| 5 | 0–0.25 | 47.79 |

*2.2. Soil Poisson's Ratio Test*

Due to the uneven shape of soil particles, it is difficult to measure the Poisson's ratio by conventional test methods. In this paper, the soil Poisson's ratio was obtained using the SINOTEST DNS02 electronic universal testing machine (as shown in Figure 2) to compress the tested 10 randomly selected soil samples.

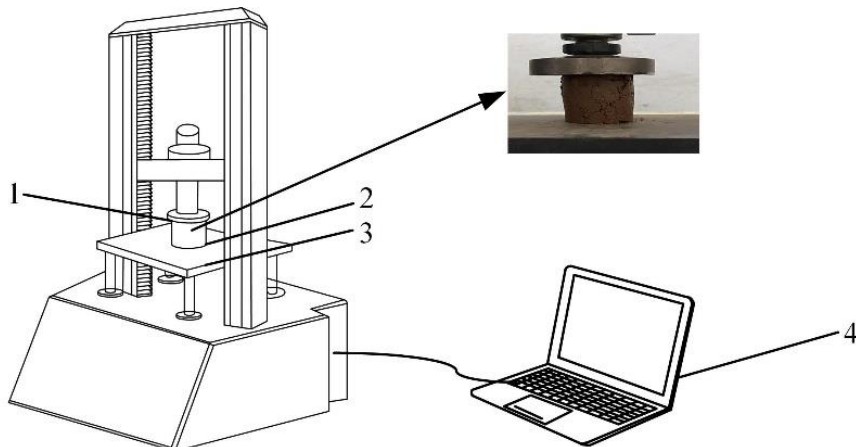

**Figure 2.** Soil Poisson's ratio test bench. (1) Circular indenter; (2) soil sample; (3) steel plate; (4) computer.

Before the test, the original dimensions of the soil sample, such as width and thickness, were measured and recorded. In the soil compression deformation test, we set the loading speed as 0.5 mm/s and applied pressure along the thickness direction of the soil sample until the soil sample cracked. When the limit cracking of the soil sample occurred under the action of axial load, we used a vernier caliper to record the deformation in the width direction and thickness direction of the soil sample. The average soil Poisson's ratio calculated by Formula (1) was 0.311.

$$\mu = \frac{|\varepsilon_x|}{|\varepsilon_y|} = \frac{\Delta L/L}{\Delta H/H} \tag{1}$$

where $\mu$ is the Poisson's ratio; $\varepsilon_x$ and $\varepsilon_y$ designate the transverse soil strain and longitudinal strain, respectively; $\Delta L$ is the absolute deformation of soil in the width direction, mm; $L$ is the original soil width, mm; $\Delta H$ is the absolute deformation of soil in the thickness direction, mm; $H$ is the original soil thickness, mm.

### 2.3. Soil Elastic Modulus and Shear Modulus Determination Test

Elastic modulus is a standard of a material's ability to resist elastic deformation. The shear modulus is usually calculated from the elastic modulus. During the test, the thickness ($H$) of the soil sample before compression was measured by a vernier caliper, and then the soil samples were placed naturally on the platform of the SINOTEST DNS02 electronic universal testing machine, which was loaded with a circular indenter at a loading speed of 0.1 mm/s. The force ($F$)-deformation ($\Delta H$) data were read, and we repeated the above tests on 10 randomly selected soil samples. According to Formulas (2)–(4), the average value of elastic modulus is 10 MPa, and the average value of shear modulus is 3.814 Mpa.

$$E = \left(\frac{F}{A}\right)/\varepsilon \tag{2}$$

$$\varepsilon = \lim_{L_1 \to 0} \left(\frac{\Delta H}{H}\right) \tag{3}$$

$$G = \frac{E}{2(1+\mu)} \tag{4}$$

where $E$ is the elastic modulus, MPa; $F$ is the axial load on the soil sample, N; $\varepsilon$ is the soil strain; $A$ is the contact area, mm$^2$, where the average contact area between the circular indenter and the soil sample is 1983.179 mm$^2$; $G$ is the soil shear modulus, MPa.

### 2.4. Soil Stacking Test

Soil stacking angle is often used to calibrate the discrete element properties of soil [18]. The cylinder lifting method was used to measure the soil stacking angle, and the measured test device is shown in Figure 3. The measuring device was composed of the SINOTEST DNS02 electronic universal testing machine, a steel cylinder (diameter 60 mm), a square steel plate base, and other devices. Before the test, a quantitative amount of loose soil was placed in a steel cylinder fitted with the contact surface of the steel plate. Then, slowly, we lifted the steel cylinder (lifting speed 0.05 m/s) until all the soil fell on the steel plate base to form a conical mound. When the soil particle slope was stable, we used a camera to take vertical photos of the front and side of the conical mound [19]. To reduce the error of manual measurement, firstly, the stacking angle graph (Figure 4a) obtained from the test was processed by grayscale processing (Figure 4b) and binarization processing (Figure 4c) using MATLAB; afterward, we transformed the edge contour curve (Figure 4d) into coordinate data through Origin software for linear fitting; and finally, the soil stacking angle was determined according to the slope of the fitting line (Figure 4e) [20]. Through 10 calculations, the average value of soil stacking angle was 27.69°.

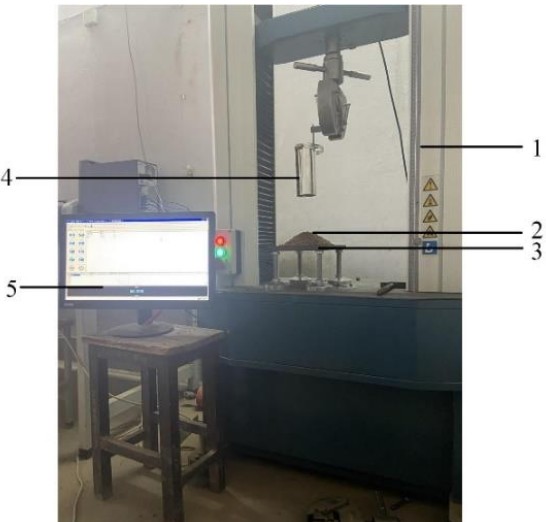

**Figure 3.** Physical test of stacking angle. (1) Universal tensile testing machine; (2) soil sample; (3) steel plate; (4) steel cylinder; (5) computer.

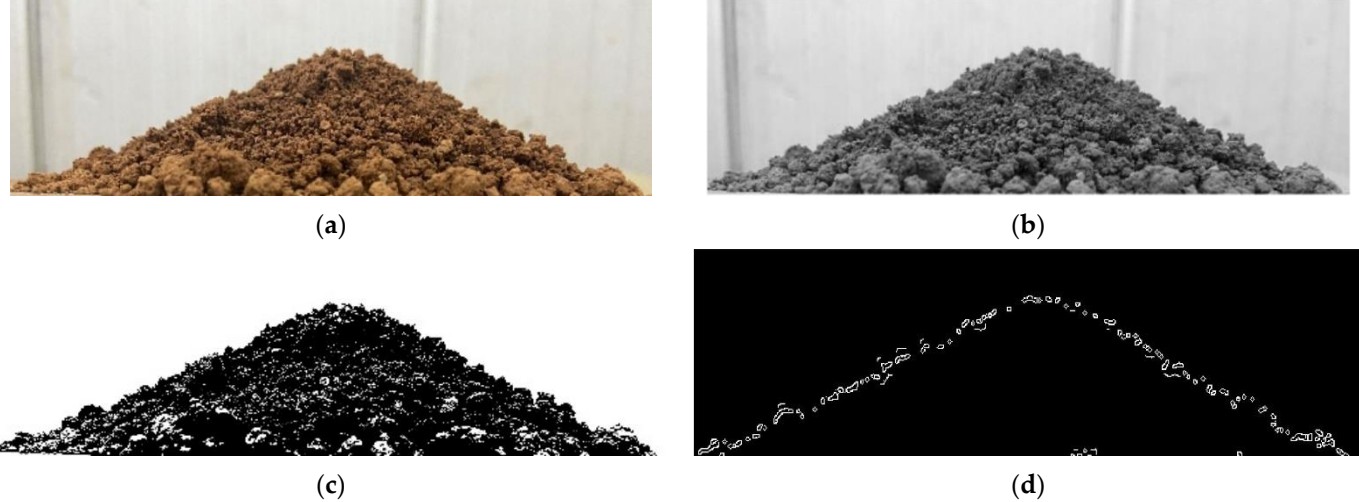

| | |
|---|---|
| (a) | (b) |
| (c) | (d) |

**Figure 4.** *Cont.*

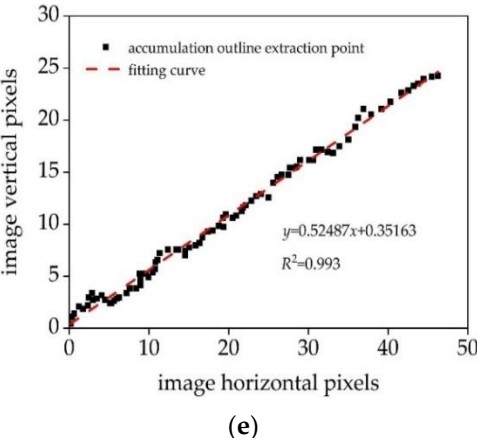

(**e**)

**Figure 4.** The calculation process of soil stacking angle. (**a**) The original image; (**b**) gray-scale image; (**c**) binarization of the image; (**d**) contour curve image; (**e**) fitting line.

## 3. Simulation Calibration of Soil Simulation Parameters

### 3.1. Soil Simulation Model

The soil accumulation process was simulated in EDEM2020 software (EDEM2020, Tory, MI, USA). As the test soil was cinnamon soil with cohesiveness, Hertz–Mindlin with JKR Cohesion model was selected as the contact model. On the basis of Hertz theory, the contact model considers the influence of the adhesion force between wet particles on the particle motion law and is suitable for simulating the materials that are bonded and agglomerated between particles due to static electricity, moisture, and other reasons, such as soil and crops [21].

The measurement of intrinsic soil parameters was completed above. On the basis of the measured soil material parameters, the particle size of the simulated soil was set to be 3 mm, 2 mm, and 1 mm, with the mass percentage corresponding to 8.85%, 8.2%, and 82.95%, respectively. The value range of each contact parameter shown in Table 2 was determined by the value of each contact parameter in the references [15,16,18,22], which were represented by $X_1$–$X_7$, correspondingly, and four virtual parameters are set, which were represented by $X_8$–$X_{11}$. The high and low levels of calibration parameters of each generation are also given in Table 2, which were represented by codes −1 and +1, respectively. The material of the test device was steel, with a density of 7850 kg/m$^3$, Poisson's ratio of 0.3, and a shear modulus of $7.9 \times 10^{10}$ MPa [23].

**Table 2.** Parameters of the Plackett–Burman test.

| Symbol | Parameter | Low Level (−1) | High Level (+1) |
|:---:|:---:|:---:|:---:|
| $X_1$ | Cinnamon soil–cinnamon soil recovery coefficient | 0.15 | 0.75 |
| $X_2$ | Cinnamon soil–cinnamon soil static friction coefficient | 0.2 | 0.92 |
| $X_3$ | Cinnamon soil–cinnamon soil rolling friction coefficient | 0.05 | 0.2 |
| $X_4$ | Cinnamon soil–steel recovery coefficient | 0.2 | 0.7 |
| $X_5$ | Cinnamon soil–steel static friction coefficient | 0.3 | 0.9 |
| $X_6$ | Cinnamon soil–steel rolling friction coefficient | 0.05 | 0.2 |
| $X_7$ | Surface energy of soil for JKR model/(J·m$^{-2}$) | 0.1 | 0.8 |
| $X_8$–$X_{11}$ | Virtual parameters | - | - |

### 3.2. Simulation Parameter Calibration Method

The moisture content, density, and other soil physical parameters in the actual agricultural soil are serious inhomogeneities. Therefore, it is necessary to accurately calibrate and optimize the parameters to be calibrated according to the recommended range. Referring to the test design in reference [24,25], aiming at the stacking angle, firstly, according to the significant parameters selected in the Plackett–Burman test, the steepest climbing test was carried out to enter the area near the optimal value quickly; then, on the basis of results of the steepest climbing test, the response surface method was used to design the Box–Behnken test, and the regression model between the stacking angle and the significant parameters was established. Taking the actual soil stacking angle as the response value, the regression model was optimized, and the optimal values of three significant parameters were obtained.

### 3.3. Results and Discussions

#### 3.3.1. Plackett–Burman Test

Table 3 shows the design scheme and simulation results of the Plackett–Burman test. The variance analysis of the simulation test results was carried out by using Design-Expert software, and the effect of each parameter on soil stacking angle is shown in Table 4.

**Table 3.** Design and results of the Plackett–Burman test.

| Number | Simulation Test Factors | | | | | | | | | | | Test Result |
|---|---|---|---|---|---|---|---|---|---|---|---|---|
| | $X_1$ | $X_2$ | $X_3$ | $X_4$ | $X_5$ | $X_6$ | $X_7$ | $X_8$ | $X_9$ | $X_{10}$ | $X_{11}$ | Stacking Angle/(°) |
| 1 | 1 | −1 | 1 | 1 | −1 | 1 | −1 | −1 | 1 | 1 | −1 | 30.2 |
| 2 | 1 | −1 | −1 | −1 | 1 | −1 | 1 | 1 | 1 | 1 | −1 | 28.6 |
| 3 | 1 | 1 | −1 | 1 | 1 | 1 | −1 | 1 | −1 | −1 | −1 | 19.2 |
| 4 | −1 | 1 | −1 | 1 | 1 | −1 | −1 | −1 | 1 | 1 | 1 | 19.0 |
| 5 | 1 | −1 | 1 | 1 | 1 | −1 | 1 | −1 | −1 | −1 | 1 | 49.3 |
| 6 | −1 | −1 | −1 | 1 | −1 | 1 | 1 | 1 | 1 | −1 | 1 | 25.3 |
| 7 | −1 | −1 | −1 | −1 | −1 | −1 | −1 | −1 | −1 | −1 | −1 | 15.1 |
| 8 | −1 | 1 | 1 | 1 | −1 | −1 | 1 | 1 | −1 | 1 | −1 | 35.2 |
| 9 | 1 | 1 | 1 | −1 | −1 | −1 | −1 | 1 | 1 | −1 | 1 | 28.9 |
| 10 | −1 | −1 | 1 | −1 | 1 | 1 | −1 | 1 | −1 | 1 | 1 | 37.0 |
| 11 | 1 | 1 | −1 | −1 | −1 | 1 | 1 | −1 | −1 | 1 | 1 | 14.9 |
| 12 | −1 | 1 | 1 | −1 | 1 | 1 | 1 | −1 | 1 | −1 | −1 | 42.0 |

**Table 4.** Analysis of significant of parameters in the Plackett–Burman test.

| Parameter | Sum of Squares | Contribution Degree | *p*-Value | Order of Significance |
|---|---|---|---|---|
| $X_1$ | 13.02 | 1.0 | 0.4523 | 6 |
| $X_2$ | 22.14 | 1.7 | 0.3391 | 5 |
| $X_3$ | 1017.52 | 77.98 | 0.0018 | 1 |
| $X_4$ | 39.24 | 3.01 | 0.2222 | 4 |
| $X_5$ | 105.02 | 8.05 | 0.0775 | 3 |
| $X_6$ | 0.52 | 0.04 | 0.8759 | 7 |
| $X_7$ | 107.4 | 8.23 | 0.0753 | 2 |

The results in Table 4 show that the simulation parameters $X_3$-soil particle-soil particle rolling friction coefficient (A), $X_7$-soil JKR surface energy (B), and $X_5$-soil particle-steel static friction coefficient (C) had significant effects on the soil stacking angle, and the other parameters had little impact, which was not significant. Therefore, in the subsequent steepest climbing test and Box–Behnken test, the calibration and optimization of three physical parameters A, B, and C with significant influence were carried out.

### 3.3.2. Steepest Climb Test

On the basis of the analysis of the Plackett–Burman test results, the steepest climbing test was carried out for the three physical parameters A, B, and C with significant influence. During the simulation process, the parameters with insignificant influence were taken as intermediate level values, namely, cinnamon soil–cinnamon soil recovery coefficient was 0.45, cinnamon soil–cinnamon soil static friction coefficient was 0.56, cinnamon soil–steel recovery coefficient was 0.45, and cinnamon soil–steel rolling friction coefficient was 0.125. The steepest climbing test design results (Table 5) show that in the level of the no. 3 test, the relative error value of the soil stacking angle reached the minimum. Therefore, the levels of no. 2, no. 3, and no. 4 were selected separately as low, medium, and high levels, respectively, for the subsequent Box–Behnken test and regression model analysis. The low, medium, and high levels of physical parameter A were 0.08, 0.11, and 0.14, respectively; the low, medium, and high levels of physical parameter B were 0.24 J/m$^3$, 0.38 J/m$^3$, and 0.52 J/m$^3$, respectively; and the low, medium, and high levels of physical parameter C were 0.42, 0.54, and 0.66, respectively.

**Table 5.** Design and results of the steepest climbing test.

| Number | Simulation Test Factors | | | Test Result | |
|---|---|---|---|---|---|
| | A | B | C | Stacking Angle/(°) | Relative Error of Stacking Angle/% |
| 1 | 0.05 | 0.1 | 0.3 | 30.2 | 51.97 |
| 2 | 0.08 | 0.24 | 0.42 | 28.6 | 16.58 |
| 3 | 0.11 | 0.38 | 0.54 | 19.2 | 4.3 |
| 4 | 0.14 | 0.52 | 0.66 | 19.0 | 10.51 |
| 5 | 0.17 | 0.66 | 0.78 | 49.3 | 27.12 |
| 6 | 0.2 | 0.8 | 0.9 | 25.3 | 49.15 |

Note: parameters A, B, and C were equal to parameters $X_3$, $X_7$, and $X_5$, respectively; the same below.

### 3.3.3. Box–Behnken Test and Regression Model

To find the optimal parameter combination of the significant parameters A, B, and C in the simulation test, on the basis of the results of the steepest climbing test, taking the stacking angle as the test index, and according to the Box–Behnken test principle, a three-factor and three-level test design was carried out, with a total of 17 tests. The simulation results in Table 6 were analyzed by variance analysis, and the analysis results are shown in Table 7.

**Table 6.** Box–Behnken test design scheme and results.

| Number | Simulation Test Factors | | | Test Result |
|---|---|---|---|---|
| | A | B | C | Stacking Angle/(°) |
| 1 | −1 | −1 | 0 | 25.7 |
| 2 | 1 | −1 | 0 | 29.8 |
| 3 | −1 | 1 | 0 | 27.5 |
| 4 | 1 | 1 | 0 | 30.8 |
| 5 | −1 | 0 | −1 | 24.2 |
| 6 | 1 | 0 | −1 | 28.2 |
| 7 | −1 | 0 | 1 | 28.6 |
| 8 | 1 | 0 | 1 | 29.1 |
| 9 | 0 | −1 | −1 | 28.1 |
| 10 | 0 | 1 | −1 | 30.1 |
| 11 | 0 | −1 | 1 | 31.2 |
| 12 | 0 | 1 | 1 | 33.1 |
| 13 | 0 | 0 | 0 | 27.6 |
| 14 | 0 | 0 | 0 | 26.8 |
| 15 | 0 | 0 | 0 | 26.6 |
| 16 | 0 | 0 | 0 | 26.1 |
| 17 | 0 | 0 | 0 | 26.7 |

**Table 7.** Variance analysis of regression model.

| Source | Sum of Squares | Degree of Freedom | Mean Square | F-Value | *p*-Value | Significance |
|---|---|---|---|---|---|---|
| Model | 78.71 | 9 | 8.75 | 24.94 | 0.0002 | ** |
| A | 17.70 | 1 | 17.70 | 50.48 | 0.0002 | ** |
| B | 5.61 | 1 | 5.61 | 16.00 | 0.0052 | ** |
| C | 16.25 | 1 | 16.25 | 46.33 | 0.0003 | ** |
| AB | 0.16 | 1 | 0.16 | 0.46 | 0.5210 | |
| AC | 3.06 | 1 | 3.06 | 8.73 | 0.0212 | * |
| BC | 0.0025 | 1 | 0.0025 | 0.00713 | 0.9351 | |
| $A^2$ | 2.09 | 1 | 2.09 | 5.97 | 0.0446 | * |
| $B^2$ | 24.15 | 1 | 24.15 | 68.88 | <0.0001 | ** |
| $C^2$ | 9.10 | 1 | 9.10 | 25.95 | 0.0014 | ** |
| Residual | 2.45 | 7 | 0.35 | | | |
| Lack of fit | 1.28 | 3 | 0.43 | 1.46 | 0.3515 | |
| Pure error | 1.17 | 4 | 0.29 | | | |
| Cor Total | 81.16 | 16 | | | | |

Note: "$p \leq 0.01$" means highly significant (**); "$0.01 < p \leq 0.05$" means very significant (*); "$p > 0.05$" means nonsignificant; the same below.

Table 7 shows the variance analysis results of the regression model. It can be seen that A, B, C, $B^2$, and $C^2$ had extremely significant effects on the stacking angle; $A^2$ and AC had significant effects on the stacking angle; AB and BC had no significant effects on the stacking angle. The regression model $p = 0.0002 < 0.01$, the lack of fit term $p$-value = 0.3518 > 0.05, indicating that the model was extremely significant. The determination coefficient was close to 1, indicating that the regression equation fitted well. The coefficient of variation CV = 2.10%, indicating that the reliability of the test results was high. The precision reached 20.092, indicating that the model can predict the soil stacking angle well. The regression equation between the soil stacking angle and the three significant parameters were expressed as Equation (5):

$$Y = 26.76 + 1.49A + 0.84B + 1.43C - 0.2AB - 0.88AC - 0.025BC - 0.71A^2 + 2.39B^2 + 1.47C^2 \tag{5}$$

In the case of ensuring that the model is significant, the items with insignificant effects (AB, BC) were eliminated, and the regression model was optimized. The results of the analysis of variance after optimization are shown in Table 8. The test precision of the model after optimization was 24.735, which was an increase of 4.643 compared with that before optimization, indicating that the model can better predict the relationship between the stacking angle and the three significant parameters. The new regression equation after optimization was

$$Y = 26.76 + 1.49A + 0.84B + 1.42C - 0.88AC - 0.71A^2 + 2.4B^2 + 1.47C^2 \tag{6}$$

**Table 8.** Regression model optimization analysis of variance.

| Source | Sum of Squares | Degree of Freedom | Mean Square | F-Value | *p*-Value | Significance |
|---|---|---|---|---|---|---|
| Model | 78.55 | 7 | 11.22 | 38.59 | <0.0001 | ** |
| A | 17.70 | 1 | 17.70 | 60.88 | <0.0001 | ** |
| B | 5.61 | 1 | 5.61 | 19.30 | 0.0017 | ** |
| C | 16.25 | 1 | 16.25 | 55.87 | <0.0001 | ** |
| AC | 3.06 | 1 | 3.06 | 10.53 | 0.0101 | * |
| $A^2$ | 2.09 | 1 | 2.09 | 7.20 | 0.0251 | * |
| $B^2$ | 24.15 | 1 | 24.15 | 83.06 | <0.0001 | ** |
| $C^2$ | 9.10 | 1 | 9.10 | 31.29 | 0.0003 | ** |
| Residual | 2.62 | 9 | 0.29 | | | |
| Lack of fit | 1.44 | 5 | 0.29 | 0.99 | 0.5198 | |
| Pure error | 1.17 | 4 | 0.29 | | | |
| Cor total | 81.16 | 16 | | | | |

Note: "$p \leq 0.01$" means highly significant (**); "$0.01 < p \leq 0.05$" means very significant (*); "$p > 0.05$" means nonsignificant; the same below.

### 3.3.4. Regression Model Interaction Effect Analysis

It can be seen from Table 8 that the interaction term AC had a significant effect on the soil stacking angle ($p < 0.05$). When the soil JKR surface energy (B) was 0.38 J/m$^{-2}$, the Design-Expert software was used to draw the response surface of cinnamon soil–cinnamon soil rolling friction coefficient (A) and cinnamon soil–steel static friction coefficient (C) interaction (Figure 5), which can visually see the interaction effect between the two parameters. The red dots in Figure 5 are the boundary and center points, corresponding to the test numbers 5, 6, 7, 8, and 13 in Table 6. It can be seen from the AC surface that with the increase in the values of the two parameters, the soil stacking angle showed a steep and similar upward trend, indicating that these two factors had a greater and basically the same influence on the soil stacking angle.

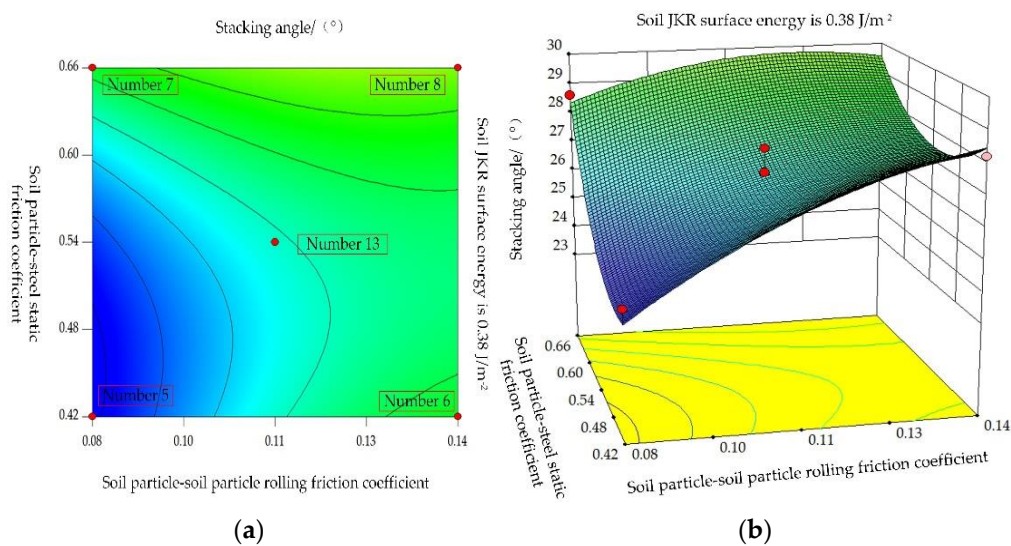

(**a**)    (**b**)

**Figure 5.** Effect of interaction on stacking angle. (**a**) Contour; (**b**) 3-D plot.

### 3.4. *Optimal Parameter Combination and Simulation Verification*

Within the range of the factor level, the Optimization-Numerical module of Design-Expert software was used to optimize the regression model of the soil stacking angle with the measured soil accumulation angle of 27.69° as the target, and several groups of optimal solutions were obtained. Through the simulation verification test, the group of solutions closest to the soil stacking angle obtained from the physical test were selected. Namely, the cinnamon soil–cinnamon soil rolling friction coefficient (A) was 0.08, the soil JKR surface energy (B) was 0.37 J/m$^{-2}$, and the cinnamon soil–steel static friction coefficient (C) was 0.64. Other non-significant parameters were taken as intermediate-level values. Using the optimized parameters to conduct three times repeated simulation tests, the average value of the simulated stacking angle was 27.62°, and the relative error with the actual physical stacking angle was 0.253%. The test comparison is shown in Figure 6, which proves the effectiveness of the simulation test.

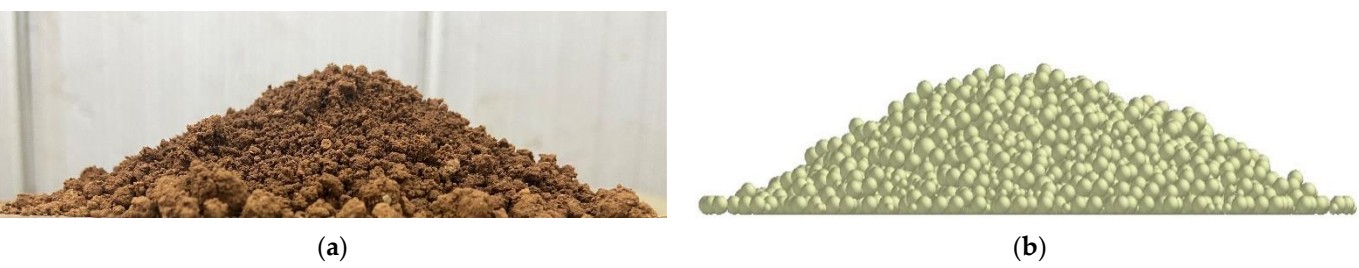

(**a**)    (**b**)

**Figure 6.** Comparisons between simulation and physical test for the soil stacking angle. (**a**) Physical test; (**b**) simulation test.

## 4. Soil Cutting Resistance Test Verification

### 4.1. Experiment Purposes

Using different soil model parameters for modeling, the macroscopic properties of soil were different, and the most obvious one was the resistance of soil-engaging components during the movement in the soil. When the bulldozing plate is used for soil cutting operations, the soil moves under the action of the bulldozing plate, which can reflect not only the interaction between the soil particle models, but also the interaction between the soil-engaging components and the soil particle models. Therefore, to verify whether the cinnamon soil discrete element simulation model constructed after parameter calibration and optimization can accurately reflect the physical and mechanical properties of the soil, the cutting resistance of the bulldozing plate during the operation was taken as the response value, compared the measured value in the soil bin test with simulation value in EDEM software, and we used the relative error value to judge the effectiveness of the cinnamon soil discrete element model.

### 4.2. Experiment Method and Index

4.2.1. Cutting Resistance Soil Bin Test

Figure 7 shows the classic structure parameters of a bulldozing plate, which mainly include the soil-engaging surfaces directrix, the cutting angle $\delta$, the clearance angle $\alpha$, the shovel point closed angle $\beta$, the front roll angle $\beta_k$, the skew angle $\varepsilon$, and the shovel point length $S$ [26]. In a general mathematical sense, all kinds of soil-engaging surfaces are formed by scanning the generatrix along the directrix following specific motion rules. Generally, the generatrix of the soil-engaging surface of a bulldozing plate is a straight line. In addition, the primary movement rule of the generatrix along the directrix is dominated by parallel translation. The directrix of the soil-engaging surface of the bulldozing plate has multiple forms. The macroscopic geometrical structure and mechanical properties of the soil-engaging surface are mainly controlled by its directrix form. The change in the directrix form will affect the contact and friction state of the soil–tool interface, the disturbance and crushing state of the soil in front of the soil-engaging surface, and the compaction state of the soil at the bottom, and will finally affect the overall cutting resistance. Therefore, the directrix form has an essential influence on the working characteristics of the bulldozer plate [27]. In engineering applications, the soil-engaging surface of a bulldozing plate is mostly a simple geometric form composed of arcs and straight lines. However, agricultural soil tillage components, such as moldboard plows, often use parabolic soil-engaging surfaces and achieve good resistance reduction effects [28]. In a previous study, the research group conducted relevant research on the bionic curve with the typical bending directrix form carried out. The bionic curve was taken from the inside contour of the longitudinal section of the vole's clawed toe with variable curvature geometric characteristics (Figure 8). Therefore, this paper carried out relevant research on three typical bending directrix forms: an arc, a parabola, and a bionic curve, and introduced the ruled surface bulldozer plate for comparison to further study the law of macro-bending geometric configuration of a soil-engaging surface on the resistance reduction performance of a bulldozer plate.

The design of the model bulldozing plate used in the test is mainly based on the fixed ones. Referring to similarity theory [29], that is, in this paper, only the directrix form was used as the only variable, and other factors were kept unchanged as much as possible, as shown in Table 9. The reference value of the front roll angle $\beta_k$ was 70°, but it still had different values, as shown in Table 9. Such a phenomenon is formed naturally after the directrix and cutting angle parameters are given; in other words, it is a dependent parameter. The cutting angle of the ruled surface was directly adopted as the 78° skew mounting angle unified by each directrix to have a unified reference to show the bending degree of several curved directrixes and to complete the comparative study of the mechanical properties of various curved soil-engaging surfaces. The height and breadth of four bulldozing plate models were set as 150 × 300 mm. The ratio of the height and breadth was amplified properly to prevent the soil from falling to the back side of the bulldozing plate from the

upper edge of the bulldozing plate during the test process because of the downsizing design of the bulldozing plate model. Otherwise, the sensor was arranged behind it, and the test precision will be affected. In addition, it is also considered that the cutting edge of the bulldozing plate has a specific linear segment, and the length S of the cutting edge of the bulldozing plate was set as 20 mm. The material of the bulldozing plate was Q235 steel. Figure 9a shows the sequence from left to right of the physical model of the bulldozing plate for the test, which is consistent with the serial numbers of 1–3 in Table 9. Figure 9b is the model photo of the ruled surface bulldozing plate added this time. Clearance angle $\alpha$ was 20°, and shovel point closed angle $\beta$ was 48°.

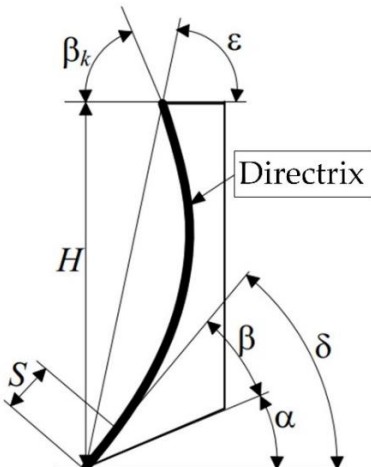

**Figure 7.** Longitudinal section structure parameters of the bulldozing plate.

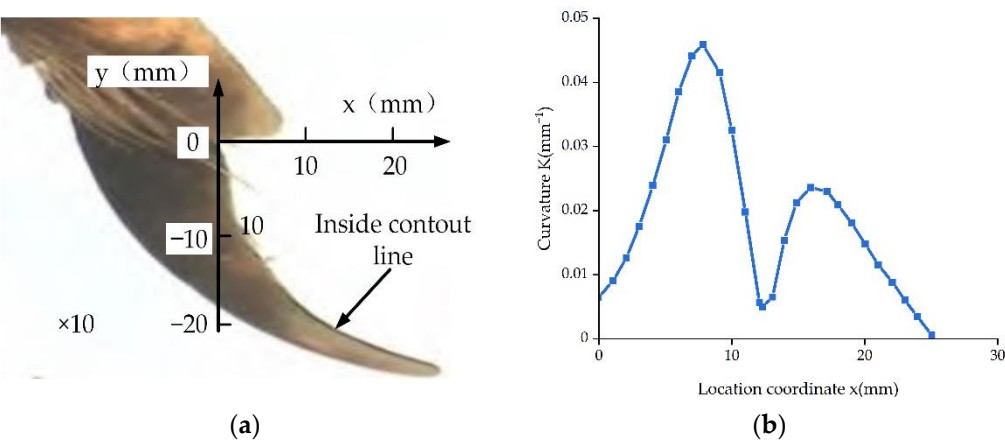

(**a**)                    (**b**)

**Figure 8.** Inner-contour line geometric features of vole claw toe. (**a**) Photograph of the middle toe of a vole's right front claw; (**b**) the curvature change trend line of the inner contour.

**Table 9.** Structure parameters of soil-engaging surface of the test bulldozing plate.

| Number | Directrix | Cutting Angle $\delta$ | Front Roll Angle $\beta_k$ | Skew Angle $\varepsilon$ | Clearance Angle $\alpha$ | Shovel Point Closed Angle $\beta$ |
|---|---|---|---|---|---|---|
| 1 | Arc | 55° | 72.4° | 78° | 20° | 30° |
| 2 | Parabola | 55° | 80.2° | 78° | 20° | 30° |
| 3 | Bionic curve | 55° | 69.4° | 78° | 20° | 30° |
| 4 | Straight line | 78° | 102° | 78° | 30° | 48° |

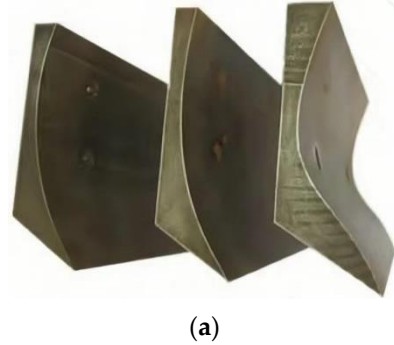 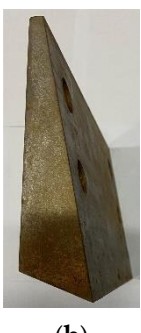

(**a**) (**b**)

**Figure 9.** Test bulldozing plate models. (**a**) Three curved bulldozing plates. (**b**) The ruled surface bulldozing plate.

Experiments were conducted to test the resistance in an indoor soil bin located in the Henan University of Science and Technology, China. The parameters of the soil bin were $6 \times 1.2 \times 0.6$ m (length × width × height). The soil type used in the test was cinnamon soil, and the soil was pretreated by deep reveling, sprinkling, and mixing before the experiment. Through these measures, it is possible to keep the soil moisture content, firmness, and other soil physical parameters consistent in each soil cutting test to enhance the comparability of test data. The trolley system was driven by a PA600 electric hoist (rated power 1.15 kW) to achieve horizontal movement through wire rope traction (traction speed 0.16 m/s). The tillage depth was controlled at 30 mm, about 20% of the height of the bulldozing plate model. The resistance signals were converted into electrical signals by the sensor and transmitted to the DH5902 data acquisition and analysis system. The data were processed by the DH5902 data acquisition and analysis system and displayed and recorded in the DHDAS signal analysis software (DHDAS, Jiangsu, China) in the computer. Figure 10a is a live photo of the test work. The bulldozing plate of the test model was connected to the trolley through three TJL-1 S-shaped force sensors. Both ends of the three S-shaped force sensors were ball-and-hinge structures, constituting a three-dimensional force system. As shown in Figure 10b, the horizontal resistance $Fx$, vertical resistance $Fy$, the resultant force $F$, and angle $\theta$ between the resultant force and horizontal force can be obtained under different working conditions by calculating the three force signals through trigonometric functions.

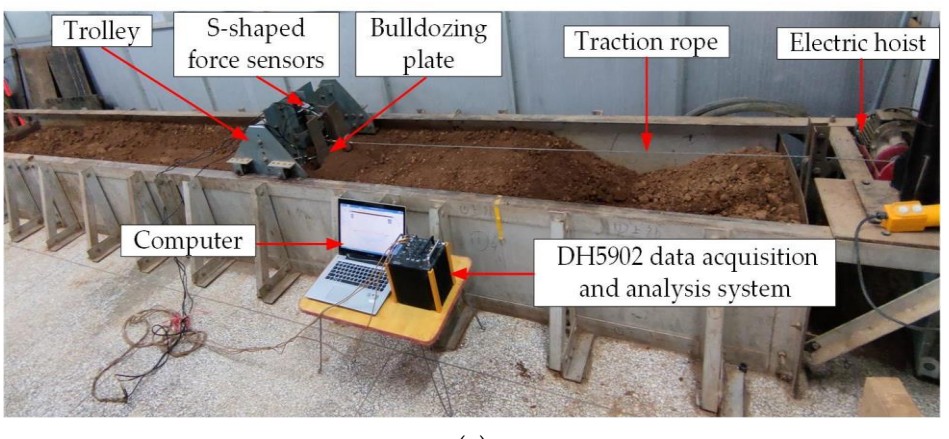 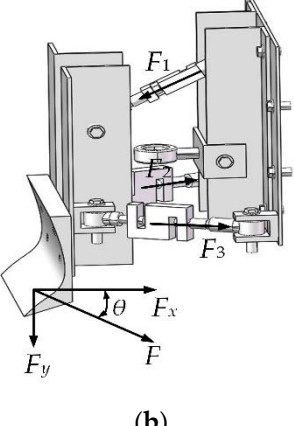

(**a**) (**b**)

**Figure 10.** The test system of soil cutting resistance. (**a**) Live photos of test work. (**b**) Schematic diagram of the force measuring device.

4.2.2. Cutting Resistance Simulation Test

Solid modeling of the bulldozing plate was carried out by using 3D drawing software Solidworks in the ratio of 1:1 (model: real object) and imported into EDEM software. The soil bin model of $1 \times 0.6 \times 0.2$ m (length × width × height) was established in EDEM

software. The soil particles were generated in the soil bin model, and the particle model and contact parameters were set to the optimal values. When the soil particles were wholly generated, the bulldozing plate advanced at a constant speed of 0.16 m/s for 4.5 s, and the tillage depth was 30 mm. The simulation test of soil cutting process and the actual test of soil cutting process are shown in Figure 11. After the simulation was over, the resistance change data during the movement were exported through the EDEM2020 software (EDEM2020, Tory, MI, USA) post-processing module.

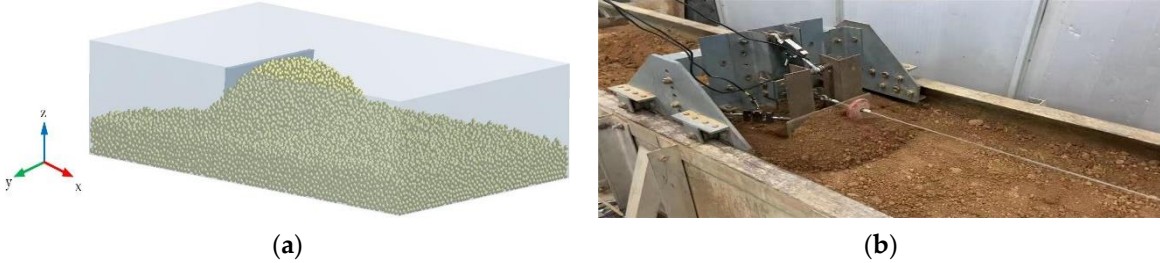

(**a**)                                                                                (**b**)

**Figure 11.** The diagram of soil cutting process. (**a**) Simulation test. (**b**) Actual test.

### 4.3. Results and Discussions

Figure 12 shows the test result of working resistance in each working condition, and the results are the average values of three repeated tests. Through comparison, it can be seen that the measured test results were slightly larger than the simulation test results because the soil conditions were relatively complex due to the existence of stones, grassroots, and other impurities in the soil during the measured test. However, the test error of the simulation test when compared with the measured test was within the acceptable range of 10.32%. It was proven that the optimal calibration of soil contact parameters is reliable and effective.

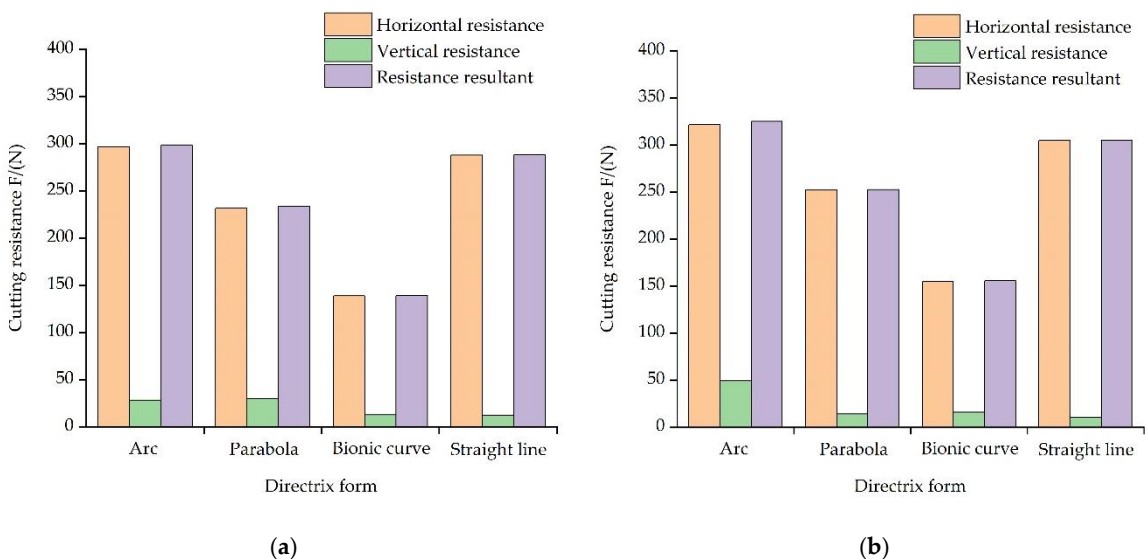

(**a**)                                                                                (**b**)

**Figure 12.** Cutting resistance of four kinds of bulldozing plates. (**a**) Simulation test. (**b**) Actual test.

Through observation of the actual test results, it can be found that the horizontal resistances of all the test bulldozing plates were significantly greater than the vertical resistances. The average horizontal working resistances of the line, arc, parabola, and bionic curve bulldozing plate were 304.80 N, 321.10 N, 252.10 N, and 154.50 N, which were 29.31, 6.58, 18.03, and 9.97 times the average vertical working resistances of 10.40 N, 48.80 N, 13.98 N, and 15.50 N, respectively. From the simulation test results, it was also found that the horizontal working resistance of the test bulldozing plate model accounted

for more than 98% of the overall working resistance, and the change trends of the two were completely similar, but the values were slightly different. Therefore, the variation law of horizontal resistance basically represents the law of bulldozing plate drag reduction. From the overall drag reduction effect, the horizontal working resistance of the bionic bulldozing plate was the smallest, the horizontal working resistance of the arc bulldozing plate was the largest, and the horizontal working resistance of the parabolic bulldozing plate was between the two. The horizontal working resistance of the circular, parabolic, and bionic surface bulldozing plates decreased by 50.60%, 17.30%, and −5.35%, respectively, compared with the ruled surface.

From the above experimental results, it can be seen that the directrixes forms of soil-engaging components had an important influence on its working resistance. When soil slides along the circular soil-engaging surface, the normal pressure borne by soil touching the soil-engaging surface points to the center. This kind of force characteristic easily makes the soil in front of the soil-engaging surface facilitate agglomeration, which is not conducive to soil fragmentation and resistance reduction requirements. Compared with the circular surface, the normal pressure generated by the ruled surface on the soil is more dispersed, which may produce relatively good soil fragmentation and resistance reduction effects. However, if the friction angle between the soil and the ruled surface is not appropriate, then soil agglomeration is easily produced, and there is a greater resistance during its working process. Different from the geometric structure and mechanical characteristics of the circular the normal pressure of the soil touching the soil-engaging surface does not point to a fixed point during the sliding process of the soil along a paraboloid with continuous variable curvature. The compacted soil will not accumulate easily, and instead, it can be crushed easily, so it may produce a lower working resistance than the circular surface in its working process. This effect will cause friction between the soil and the soil-engaging surface, and the internal stress of the soil within a large range in the front of the soil-engaging surface will produce high-speed alternations or fluctuations. This effect will lead to friction between the soil and the soil interface, and the internal stress of the soil in a large range in front of the soil interface will produce high-speed alternation or fluctuation. There are also two external manifestations of the soil cutting process with these stress characteristics. One is that there is less soil adhesion to the soil-engaging surface, and the disturbed soil has a good crushing effect. Second, the overall work resistance is low.

## 5. Conclusions

(1) Aiming at the cinnamon soil with yellow sandy soil characteristics commonly found in wheat fields in Luoyang, China, the Hertz–Mindlin with JKR as the contact model between soil particles, and the soil samples with a moisture content of 11.62% were simulated by using the method of combining physical test and simulation test. Through tests, we found that the average Poisson's ratio of soil was 0.311. The average elastic modulus of soil was 10 MPa. The average shear modulus was 3.814 MPa. The mean value of the soil stacking angle was 27.69°.

(2) On the basis of the physical test results, the Plackett–Burman test was carried out. The results of variance analysis showed that the cinnamon soil–cinnamon soil rolling friction coefficient (A), soil JKR surface energy (B), and cinnamon soil–steel static friction coefficient (C) had significant effects on the soil stacking angle.

(3) The Box–Behnken test was carried out on the significant parameters, and the second-order regression model of the accumulation angle and the significant parameters was established. Taking the accumulation angle test value of 27.69° as the objective, the optimal solution was obtained: cinnamon soil–cinnamon soil rolling friction coefficient (A) was 0.08, soil JKR surface energy (B) was 0.37 J/m$^{-2}$, and cinnamon soil–steel static friction coefficient (C) was 0.64. The relative error between the optimal parameter combination stacking angle and the actual physical stacking angle was 0.253%.

(4) The working resistance test data were consistent with the analysis results, which showed that the directrixes forms of soil-engaging components has an important

influence on its working resistance. Horizontal working resistance of the circular, parabolic, and bionic surface bulldozing plates decreased by 50.60%, 17.30%, and −5.35%, respectively, compared with the ruled surface. Exploring the influence mechanism between the directrix form of soil-engaging surface and working resistance in order to obtain lower working resistance will be one of the main research directions of energy-saving design of soil contact surface in the future.

(5) To verify the accuracy of the calibrated and optimized parameters of the discrete element model, the indoor soil bin cutting resistance test and the simulation cutting resistance test were compared and analyzed. It was found that the test error of the simulation test compared with the measured test was within the acceptable range of 10.32%, indicating that the physical and mechanical properties of the simulation soil model were essentially consistent with the actual soil, which verified the accuracy and reliability of the cinnamon soil discrete element simulation parameter calibration results and research methods. The research results can provide basic data for the simulation of cutting resistance research of cinnamon-soil-engaging components.

**Author Contributions:** Conceptualization, Y.Q. and Z.G.; methodology, Y.Q.; software, X.J. validation, Z.G. and X.J.; formal analysis, P.Z.; investigation, S.S.; resources, F.G.; data curation, Y.Q.; writing—original draft, Y.Q.; writing—review and editing, Y.Q.; visualization, Y.Q.; supervision, Z.G.; project administration, Z.G.; funding acquisition, Z.G. All authors have read and agreed to the published version of the manuscript.

**Funding:** This research was funded by the National Nature Science Foundation of China, grant number 51675163 and 51175150.

**Institutional Review Board Statement:** Not applicable.

**Informed Consent Statement:** Not applicable.

**Data Availability Statement:** Not applicable.

**Conflicts of Interest:** The authors declare no conflict of interest.

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
