# Peer review of "Calibration and Verification Test of Cinnamon Soil Simulation Parameters Based on Discrete Element Method"

_agriculture, doi:10.3390/agriculture12081082_

Round 1

Reviewer 1 Report

Line 43 - specify home (put the name of the place and country)

The paper is well written and well prepared. Just proofread the whole manuscript and fix some sentences  e.g. the last sentence in conclusion (Line 394)

What about the soil composition? It would be important to do XRF and PXRD to know the composition and phases in the soil. 

Include the best practices whereby cinnamon soil can be used and provide reasons.

The points of soil collection should also be known. Provide a map indicating the locations. Thus, when you specify that you have selected random samples, indicate where they were collected. 

Based on your results, what are the best soil-engaging components that can be used? Add this in your conclusion

Reviewer 2 Report

Correct the manuscript:

1. The relevance of the study is associated with the creation of a simulation model of cinnamon soil and its interaction with a bulldozer plate, and the novelty lies in improving the accuracy of the model and the use of plates. Except often with bionic contour. Relevance and novelty are beyond doubt. Now the manuscript lacks analysis, comparison and conclusion regarding the control (conventional) bulldozer plate;

2. In the annotation, it is necessary to add about the relevance of the study and reflect the bionic factor;

3. The interpretation of the regression equation (5) has not been fully carried out. It is necessary to explain the reasons for the influence of factors (mechanism);

4. in figures 4, you should additionally indicate the value of the third fixed factor (B) and decipher the red dots in the notation;

5. figure 10 must be supplemented with appropriate photographs of the real process;

6. In the conclusions and abstracts, add an assessment of the effectiveness of the process of interaction of soil with a bionic bulldozer plate, and also add a perspective for the development of your research.

Round 2

Reviewer 2 Report

The article has been significantly improved. The authors responded to my comments. I recommend the article for publication in this version.